# Evidence of Spaceflight-Induced Adverse Effects on Photoreceptors and Retinal Function in the Mouse Eye

**DOI:** 10.3390/ijms24087362

**Published:** 2023-04-17

**Authors:** Xiaowen Mao, Seta Stanbouly, Jacob Holley, Michael Pecaut, James Crapo

**Affiliations:** 1Department of Basic Sciences, Division of Biomedical Engineering Sciences (BMES), Loma Linda University Health, Loma Linda, CA 92350, USA; 2Department of Medicine, Division of Pulmonary, Critical Care & Sleep Medicine, National Jewish Health, University of Colorado Denver, Denver, CO 80204, USA

**Keywords:** spaceflight, retina, photoreceptor, oxidative stress, antioxidant

## Abstract

The goal of the present study was to characterize acute oxidative damage in ocular structure and retinal function after exposure to spaceflight, and to evaluate the efficacy of an antioxidant in reducing spaceflight-induced changes in the retina. Ten-week-old adult C57BL/6 male mice were flown aboard the ISS on Space-X 24 over 35 days, and returned to Earth alive. The mice received a weekly injection of a superoxide dismutase mimic, MnTnBuOE-2-PyP 5+ (BuOE), before launch and during their stay onboard the ISS. Ground control mice were maintained on Earth under identical environmental conditions. Before the launch, intraocular pressure (IOP) was measured using a handheld tonometer and retinal function was evaluated using electroretinogram (ERG). ERG signals were recorded when the mouse eye was under dark-adapted conditions in response to ultraviolet monochromatic light flashes. Within 20 h after splashdown, IOP and ERG assessments were repeated before euthanasia. There were significant increases in body weight for habitat control groups post-flight compared to pre-flight measurements. However, the body weights were similar among flight groups before launch and after splashdown. The IOP measurements were similar between pre- and post-flight groups with no significant differences between BuOE-treated and saline controls. Immunofluorescence evaluation showed increases in retinal oxidative stress and apoptotic cell death after spaceflight. BuOE treatment significantly decreased the level of the oxidative stress biomarker. ERG data showed that the average amplitudes of the a- and b-wave were significantly decreased (39% and 32% by spaceflight, respectively) compared to that of habitat ground controls. These data indicate that spaceflight conditions induce oxidative stress in the retina, which may lead to photoreceptor cell damage and retinal function impairment.

## 1. Introduction

Neuro-ophthalmic abnormalities have been identified in astronauts. These complex physiological and pathological changes are collectively defined as spaceflight-associated neuro-ocular syndrome (SANS) [1]. The condition can lead to long-term effects, including changes in visual acuity, even after the astronauts have returned to Earth. The underlying mechanisms of the changes and functional outcomes of observed ocular change remain poorly understood.

An increase in oxidative stress in the retina and photoreceptors is thought to be a key factor in the development of retinal disease [2]. Overproduction of reactive oxygen species (ROS) leads to damage of the critical cell populations and tissue remodeling. Previous studies have revealed enhanced immunoreactivity for the oxidative biomarker, 4-hydroxynonenal (4-HNE), in the retina, following space radiation exposure [3] and spaceflight [4]. 

Because environmental, radiotherapy, and pathological conditions can induce the production of a high level of unstable oxygen species, efforts have been made to develop drugs with therapeutic potential. However, practical application has been somewhat disappointing due to complications such as short half-lives, low reaction rate constants, lack of cellular uptake, and hypersensitivity reactions [5]. Metal-containing catalytic antioxidants have emerged as especially promising. Of these, redox-active metalloporphyrins possess distinct antioxidant properties, including scavenging O_2_^−^, H_2_O_2_, and lipid peroxides that mediate cellular and tissue damage [6]. These compounds have proven efficacious in numerous in vivo animal models of disease in reducing oxidative stress [7,8,9,10]. Furthermore, in previous studies, we documented the protective effects of a redox-active metalloporphyrin (MnTE-2-PyP) to both photoreceptors and retinal capillaries from radiation damage [11,12]. 

Butoxyethylpyridylporphyrin–MnTnBuOE-2-PyP5+, (BuOE), is a superoxide dismutase (SOD) mimic, that is also termed BMX-001, and is currently in multiple phase 1 and phase 2 human clinical trials. MnTnBuOE-2-PyP5+ has shown protective effects against radiation-induced damage to the brain white matter and hippocampal neurogenesis [13,14].

Our previous flight data showed a significant increase in apoptotic damage in the photoreceptors of flight mice compared to ground controls [4]. However, the pathophysiological consequences of cellular damage on photoreceptor and retinal function have not been well investigated. In the present study, mice were flown on SpX-24 aboard the International Space Station (ISS) for 35 days to characterize the space environment-induced oxidative damage on ocular structure and retinal function using histochemistry, intraocular pressure (IOP), and electroretinography (ERG). The antioxidant effects of BuOE in mitigating spaceflight condition-induced retinal damage were evaluated. 

## 2. Results

### 2.1. Body Weight

Mice were weighed before launch and shortly before being euthanized after landing. Prior to taking off, the body weights were similar among all groups. The flight (FLT) and habitat ground control (GC) groups weighed between 25 and 27 g. The body mass after landing, for both the saline control and BuOE-treated FLT mice, was not significantly different compared to that of pre-flight measurements. However, body weights for the age-matched GC groups were significantly (*p* < 0.05) increased following the flight compared to pre-flight measurements (Figure 1).

### 2.2. IOP Measurement 

IOP was measured from a randomly selected left or right eye of each mouse before launch and shortly after landing. The post-flight saline and BuOE-treated FLT groups showed a decreasing trend in IOP, compared to pre-flight measurements, but there were no statistical differences with 11.4 and 10.2 mmHg for pre- vs. post-flight saline group, and 11.3 and 10.3 mmHg for pre- vs. post-flight BuOE-treated groups, respectively. The measurements for the flight groups were similar to that of the GC groups (Figure 2).

### 2.3. ERG Assessment of Retinal Function

To assess the mouse retinal function, we measured the amplitude of the ERG a-wave, which is generated by the activation of photoreceptors, a measure of photoreceptor function, and b-wave, which is primarily generated by the ON-type bipolar cell neurons [15]. Representative ERG waveforms for FLT saline vs. GC saline and FLT saline vs. FLT BuOE-treated groups were compared in Figure 3a,b. The figures illustrate average a- and b-wave amplitude measurements in response to the strongest flash, with stimulus energies of log 2.5 cd·s/m^2^ over implicit time. The amplitudes for a- and b-waves for the flight groups were reduced when compared to controls. The waveforms were similar between post-flight saline and BuOE-treated groups. The longitudinal changes of amplitude and implicit time (time-to-peak) for a- and b-waves with increased flash intensities for pre- and post-flight FLT and GC groups were presented in Figure 4 and Figure 5. Differences in amplitude and implicit time to light stimulation were noted in the FLT group following the spaceflight when they were compared to pre-flight measurements or the GC group. Reduced amplitudes and delayed peak time observed in the FLT group may indicate low sensitivity of cone cells in response to stimulus flash. 

When compared to pre-flight, the amplitude of the a- and b-wave that reflected an average response to all flash intensities in the post-flight group was significantly reduced (*p* < 0.05) for both saline and BuOE-treated groups (Figure 6a,c), while the amplitude of a- and b-waves for pre- and post-flight in the GC group was similar (Figure 6b,d).

When the a-waves of post-flight groups were compared, there were statistical differences (*p* < 0.05) in average amplitudes between FLT and GC (Figure 7), while no differences were noted among the groups when measured pre-flight. The amplitude of the a-wave in the BuOE-treated group was close to the GC controls. Our ERG data also showed that the average amplitude of the b-wave that elicited a light-invoked slow cornea-positive potential was significantly (*p* < 0.05) decreased by 39% and 25% in FLT saline or BuOE-treated mice compared to their habitat ground control counterparts, respectively (Figure 8). No differences for b-waves were found among any of the pre-flight groups. 

### 2.4. Apoptosis in the Retina 

Apoptosis in the retina and retinal endothelial cells (ECs) was evaluated using terminal deoxynucleotidyltransferase dUTP nick-end label (TUNEL) staining. Apoptotic cells labeled with green fluorescence (Figure 9a) were mostly detected in the retinal ganglion cell layer (GCL) and inner nuclear layer (INL). Quantitative assessment revealed that the density of apoptotic cells in the retina of the flight saline group was 67% higher than GC controls, with a strong trend difference (*p* = 0.056) (Figure 9b). The counts for apoptosis were similar between the BuOE-treated and saline flight groups. The data demonstrate that spaceflight conditions induce retinal apoptosis. The retinal ECs showed a similar level of apoptosis across all groups (Figure 9c). 

### 2.5. Oxidative Damage in the Photoreceptor

The 4-hydroxynonenal (4-HNE) marker for oxidative damage was evaluated in the retina. Enhanced 4-HNE staining was seen in cone photoreceptors, retinal INL, and GCL in the spaceflight group compared to all other groups (Figure 10a). As shown in Figure 10b, the fluorescent intensity of 4-HNE in the photoreceptor cones of the flight group mice was significantly increased (*p* < 0.05) compared to both the ground controls and the BuOE-treated flight group. The immunoactivity of HNE immunoreactivity in BuOE-treated FLT retina was similar to saline or BuOE-treated GC groups. 

Cone photoreceptors were stained with a specific immunofluorescent marker of peanut agglutinin (PNA). Quantitative evaluation (Figure 10c) revealed a decrease of 20% in the cone photoreceptor density in the spaceflight saline mice, with an average of 812 counts/mm^2^ compared to the GC group which had an average of 1007 counts/mm^2^ (*p* = 0.067). There were no significant differences in the cone density between the flight saline and BuOE groups.

## 3. Discussion

Observed changes following spaceflight include increased oxidative damage and increased apoptosis in the mouse retinal nuclear layer and photoreceptors. ERG data, for the first time, demonstrated that spaceflight evoked changes in photoreceptor function in a mouse model. The average amplitudes of a- and b-waves for the flight group were significantly reduced when compared to controls.

Our results show that FLT mice did not gain weight after the flight as was observed for the age-matched GC group. Other spaceflight studies reported a loss in body mass after spaceflight [16,17]. It has been proposed that the lack of increase in body mass may be related to mixed factors, including dehydration, stress, altered glucose, and lipid metabolism, or less food consumption [18,19]. 

Elevated IOP is a major risk factor for the development and progression of many ocular diseases, especially glaucoma [20]. IOP and intracranial pressure (ICP) changes are suspected to be involved in the ocular anatomical changes and visual function impairment exhibited by some astronauts [21]. Results from head–down bed-rest studies, which simulate the microgravity effects, have shown IOP and ocular blood flow changes. These changes are implicated in the pathophysiology of optic disk edema [22]. For our study, within 20 h of splashdown, we measured the IOP of flight and control groups in fully conscious mice. The data were consistent with previous observations and showed no significant differences between pre- and post-flight groups, or between flight and control groups. This indicates that a 35-day space mission may have a limited impact on irreversible IOP changes pathologically. One possible explanation is that IOP is a transient and time-sensitive parameter. Changes can happen within seconds to minutes. Due to logistical limitations, we were unable to measure IOP until 20 h after splashdown, and elevations in this readout may not have been detectable. Studies have shown that IOP fluctuates considerably with time, body postural change, venous pressure changes, and stress levels [23]. Further evaluation of IOP during spaceflight or longer time sustained microgravity conditions will be required.

The ERG is an electrical potential generated in response to light stimulation of the retina and recorded from the corneal surface of the eye. ERG technology is a powerful approach in assessing retina function [24] and ganglion cell activity [25]. The mouse retina expresses a short wavelength-sensitive and a middle/long wavelength-sensitive opsin (S- and M-opsin) [26], whereby the S-cones are responsive to ultraviolet (UV) peak at 360 nm, and M-cones and rods are responsive to the green light at peak 504 nm. Previous studies suggested that the mouse cone system predominantly expresses S-opsin [27]. Since S-opsin sensitivity drops significantly at wavelengths longer than 425 nm, only UV light is efficient in stimulating mouse S-opsin [27]. In our present study, we focused on probing S-cone-mediated properties using UV light stimulation. Despite the small percentage of cones in the mouse photoreceptor (5%) [28], cone-mediated responses in mouse ganglion cells are fast and sensitive. Our measurement enables us to quantify cone-mediated response and correlate this with histochemistry evaluation. Cone bipolar pathways are similar to those found in mammals [29]. Therefore, the mouse is potentially a useful model to study cone-mediated vision and for diseases that affects the cone function in humans [27,29].

The two ERG components that are most often measured are the a- and b-waves. The a-wave is the first large negative component that reflects the physiological status of photoreceptor rods and cones, followed by the b-wave, which is a cornea-positive amplitude. The b-wave reflects the health of the inner layers of the retina, including the ON bipolar cell and Müller cell depolarization in response to increases in light stimulation [15]. Reduced a- and b-wave amplitudes documented in the flight group indicate retinal photoreceptor damage and disturbance of retinal physiological function. These changes are associated with the observed increase in apoptosis in the inner nuclear ganglion cell layer, and a decrease in cone density following spaceflight. Abnormal ERG is recognized in many ocular diseases, including diabetic retinopathy, retinitis pigmentosa, glaucoma, and cone dystrophy [15,30,31]. Most human retina disorders are detected by attenuation of amplitude [25]. Studies have also shown a correlation between a decrease in cone-driven function through reductions of dark-adapted ERG b-wave amplitudes and cone photoreceptor loss in a light damage model [32]. Our ERG and histological findings, for the first time, show that spaceflight-induced retinal functional change may occur as a result of photoreceptor damage and cellular death. It is important to characterize and monitor retinal changes that may lead to the progression of retinal disease and retinal degeneration. 

Retinal neurons include amacrine, ganglion cells, and photoreceptors. Radiation-induced photoreceptor loss has been shown in a previous study with 1 Gy of iron-56 irradiation [33]. Studies from the Space Shuttle Mission (STS-135) and Rodent Study-9 (RR-9) also showed a significant increase in apoptotic cell damage in the outer nuclear layer (ONL) and INL of the flight mice photoreceptors [34,35]. Retinal nuclear cell damage causes irreversible blindness in many retinal diseases [36]. Retinas are more sensitive to environmental damage than other central nervous system (CNS) neurons [37]. Apoptotic cell death is documented in many forms of retinal vascular dysfunction and degeneration [38]. These findings suggest that retinal nuclear and photoreceptor cells play an important role in maintaining the normal structure and function of the retina. Our data revealed significant increases in retinal nuclear/photoreceptor cell death and a trend showing a decrease in cone density by spaceflight. The presence of apoptotic positive cells in this study supports the hypothesis that apoptotic mechanisms are involved in retinal cell death. Further studies will be required to determine whether these effects are reversible or continue to deteriorate over time. 

Oxidative stress-induced ocular tissue damage has been associated with a variety of pathological conditions [2,39,40]. Oxidative stress is one of the leading causes of retinal damage and retinal degenerative diseases [41]. Radiation-induced oxidative stress can damage critical cellular structure and biomolecules [42], resulting in inflammatory responses, cell death, and tissue remodeling [43,44]. We have shown in our previous studies that spaceflight and space radiation exposures cause oxidative changes in the retina and retinal vasculature [34,45,46]. HNE is a secondary product of lipid peroxidation during oxidative stress [47]. This flight study documented increased oxidative damage with 4-HNE staining, a biomarker for lipid peroxidation in cone photoreceptors, retinal INL, and GCL [4]. The correlation between increased oxidative damage in photoreceptors and abnormal ERG recording indicates that spaceflight impacts the retinal structure, physiology and function. In future studies, we will determine if these observed changes will persist or reverse with a longer re-adaption period.

The safety of BuOE has been evaluated in a battery of nonclinical Good Laboratory Practice (GLP)-compliant studies. Systemic toxicity has been evaluated using the intended cGMP product administered subcutaneously for periods of up to 5 weeks in both mouse and the monkey, and have included toxicokinetic evaluations to characterize systemic exposure, tissue distribution, and clearance of BuOE [48]. Redox-active metalloporphyrin antioxidants have been shown to be effective in protecting against radiation-induced ocular damage [11]. In the present study, with the current injection dose, our results showed that BuOE is safe. Body weights among saline- or BuOE-treated ground control groups were similar and no overt sign of toxicity related to this compound was observed. 

Immunoreactivity of the oxidative stress biomarker, HNE, was significantly reduced in the BuOE-treated flight group compared to the flight saline group. This suggests that BuOE was effective in reducing reactive oxygen species that activate retinal stress responses during space flight [49,50]. Metalloporphyrin catalytic antioxidants are potent detoxifiers of lipid peroxides and lipoperoxyl radicals, a process that is mediated via hydroxyl radicals [6,51]. Our present study suggests that oxidative stress is a contributing mechanism for observed damage in the retina and photoreceptor, which was demonstrated by increased 4-HNE staining, a biomarker for lipid peroxidation. Adding BuOE, a lipophilic metalloporphyrin compound with high potency for the catalysis of lipid peroxidation reduced immunoreactivity of the biomarker in the BuOE-treated ocular tissue compared to the flight saline group, suggesting a potential mechanism of oxidative stress in inducing cellular damage.

However, BuOE was found to only slightly mitigate spaceflight-induced apoptosis in retinal photoreceptor or endothelial cells, suggesting that apoptosis was not inhibited. Administration of BuOE had limited effects in attenuating spaceflight-induced retinal functional change assessed by ERG recording, although it was more effective in restoring a-wavet responses than b-wave responses, especially at high flash intensity (Figure 3b). It may be that the beneficial or protective effects of BuOE with the current dosage was not sufficient to significantly ameliorate the harmful spaceflight-induced effects on the retina or that alternative stressors may be directly leading to apoptosis despite reduced oxidative stress. Due to logistic limitations, the compound was injected once a week in orbit. More frequent injections may be required to protect retinal function more effectively. Daily [52,53] or twice-weekly [54] administration schedules have been reported to have significant protective effects from oxidative damage in other disease models. Free radicals produced in the ocular tissues by the mechanism other than ROS, which include reactive nitrogen species, and reactive carbonyl species [55], may be equally important in inducing cellular damage in the retina. Another plausible explanation is that spaceflight conditions induced an imbalanced oxidative/antioxidative homeostasis and antioxidant defense enhanced by this redox-active compound may lead to apoptotic pathway regulation and recovery of retinal function at a later time. We plan to evaluate oxidative stress biomarkers, apoptotic status, and retinal function parameters at three months following live animal return to Earth, to further assess the ocular tissue re-adaptation process. It is anticipated that a redox-active metalloporphyrin can be effective in protecting against spaceflight-induced retinal damage.

## 4. Material and Methods

### 4.1. Spaceflight and Mouse Groups

Ten-week-old C57BL/6 male mice were launched in December 2021, at the Kennedy Space Center (KSC) on the rodent research-18 (RR-18) mission for 35 days aboard the ISS. All mice were maintained at an ambient temperature of 26–28 °C with a 12 h light/dark cycle during the flight. This hardware has a housing density that is within the guidelines recommended by the National Institutes of Health. All mice were provided NASA Nutrient-upgraded Rodent Food Bar (NuRFB) and autoclaved deionized water ad libitum. The flight mice were subdivided into two groups: a saline control or BuOE-treated group. BuOE at 1 mg/kg (0.2 mL) was administrated subcutaneously 7 days prior to the flight launch and weekly aboard the ISS. After returning to Earth via SpaceX’s Dragon capsule, it took approximately 20 h for the mice to be recovered in the Atlantic Ocean, brought to shore, and transported to the research laboratory at Roskamp Institute, Sarasota, Florida. Upon live return, a subset of mice (8 per group) went through IOP and ERG testing. The rest of the spaceflight mice (5 per group) were then euthanized with intraperitoneal (IP) injections of Ketamine/Xylazine (up to 150 mg/kg Ketamine combined with up to 45 mg/kg Xylazine). Their eyes were removed and prepared for analysis. The left eyes were fixed in 4% paraformaldehyde in phosphate-buffered saline (PBS) for 24 hours, and then rinsed with PBS for immunohistochemistry (IHC) assays. The right eyes were dissected to obtain the retina. The retinas were flash-frozen and stored at −80 °C for further analysis. Ground control mice were placed into the same housing hardware with environmental parameters used in flight. GC mice were euthanized three days later. Animal experiments were approved by the National Aeronautics and Space Administration (NASA) Animal Care and Use Committee (IACUC) on 14 October 2021 (Protocol Number: RR-18), Roskamp Institute IACUC on 7 October 2021 (Protocol Number RR-18), and Loma Linda University Health IACUC on 8 February 2022 (Protocol Number: 8170051). The activities involving vertebrate animals were conformed to the Guide for the Care and Use of Laboratory Animals (National Institutes of Health, Bethesda, MD, USA).

### 4.2. IOP Measurement 

IOP in conscious animals was measured pre-launch and post-flight before euthanization with a handheld tonometer (TonoLab-ICARE, Raleigh, NC, USA). The animals exhibited no signs of discomfort during the procedure. Five to six sequential measurements were made to obtain an average of the counts as a single IOP readout for each mouse [56]. 

### 4.3. ERG

To measure photoreceptor function, ERG was recorded using a Micron^®^ Ganzfeld ERG system (Phoenix-Micron, Inc., Bend, OR, USA), according to the manufacturer’s instructions. Mice were dark-adapted for at least 3 h prior to obtaining ERG recordings which were performed on the right eyes. Mice were maintained under anesthesia with 1.5–2% Isoflurane through a nose cone and placed on a heated stage during testing to prevent hypothermia throughout the procedure. The pupils were dilated with 1–2 drops of 0.5% tropicamide (Akorn, Inc. Lake Forest, IL, USA) and 0.5% phenylephrine hydrochloride (Paragon BioTeck, Inc. Lake Forest, IL, USA) ophthalmic solutions. Hypromellose (Gonak™) ophthalmic demulcent solution (2.5%) (Akorn, Inc.) was topically administered to the eyes for proper contact with the corneal electrode. The Ganzfeld optical head includes three LEDs with 360, 504, and 850 nm wavelengths. It illuminates the eye with near-infrared at 850 nm during alignment of the corneal electrode guided by the Eye-Cockpit ( IDS uEye software version 4.3, IDS Imaging Development Systems Inc., Minato City, Tokyo) camera software. Three electrodes were attached to record the transretinal electrical signals: the Ganzfeld corneal (positive) electrode was made of a gold conducting ring around the objective lens, contact-attached to the cornea, sealing the eye. A reference (negative) electrode was inserted subcutaneously through the head skin positioned midline between the ears, such that the end of the needle was between the eyes. Finally, a ground electrode was inserted subcutaneously through the base of the tail. ERG was recorded according to the manufacturer’s protocol. Following signal baseline stabilization, ERG signals were recorded in response to increasing intensities of 360 nm Ultraviolet monochromatic light flashes (1 ms duration). Ganzfeld uses a Maxwellian view of illumination, which projects single light to the eye pupil illuminating the retina with high divergence. For each mouse, five sweeps of electrical responses were acquired at each intensity flash, with varying stimulus intervals (Table 1). The waveform amplitudes (microvolts) and implicit times (seconds) of a- and b-waves at each flash stimulus were acquired from five averaged sweeps and analyzed using the Labscribe software (LabScribeERG version 3.016800, iWorx systems, Dover, NH, USA). The amplitude of the a-wave represents the photoreceptor layer hyperpolarization (negative) signal and the b-wave represents the inner retinal bipolar cell and Müller cell depolarization (positive) response signal. Data obtained from all mice were averaged to represent the mean ± standard error (SEM) of a-wave and b-wave amplitudes.

### 4.4. TUNEL Assay

Six μm sections were cut through each left eye for analysis. Five ocular sections, roughly 100 μm apart, were evaluated using the TUNEL assay to identify and quantify apoptotic cells, according to standard procedures. Briefly, paraffin-embedded sections were processed using DeadEnd™ Fluorometric TUNEL system kit (catalog no. G3250, Promega Corp., Madison, WI, USA). The same sections were then incubated with DyLight 594 Lycopersicon esculantum-Lectin (1:100, catalog no. DL-1177, Vector laboratories Inc., Burlingame, CA, USA) to stain the endothelium. Nuclei were counterstained with diamidino-2-phenylindole (DAPI, blue). 

### 4.5. IHC Staining

Six µm paraffin-embedded section of the left eye from 5 mice per group, roughly 100 μm apart, were used for analysis. To evaluate oxidative damage in the retina, immunostaining was performed on ocular sections using 4-HNE antibody specific for lipid peroxidation. For double labeling of HNE and PNA, sections were incubated with fluorescein isothiocyanate (FITC)-conjugated peanut agglutinin (catalog no. FL-1071, Vector laboratories Inc., Burlingame, CA, USA) (1:100 in 1% BSA) for an hour at room temperature, for labeling the photoreceptor cones. Sections were then incubated overnight (18–21 h) at 4 °C with rabbit primary anti-4-HNE antibody (1:1000, catalog no. HNE11-S, Alpha Diagnostic Intl. Inc. San Antonio, TX, USA). After washing three times in PBS, sections were further treated with secondary antibody Alexa Fluor 568 goat anti-rabbit IgG (catalog no. A11011, Life Technologies Corp., Eugene, OR, USA). The cell nuclei were counterstained with DAPI, and were mounted and cover-slipped with Vectashield Vibrance Antifade Mounting Medium (catalog no. H-1700, Vector Laboratories Inc., Burlingame, CA, USA).

### 4.6. Quantification of Immunostaining

Six to 10 field images were examined using a BZ-X710 All-in-One inverted fluorescence microscope with structural illumination at 20X magnification, spanning the entire retina per section. For quantitative analysis, the total numbers of TUNEL-positive nuclear /ganglion cells or endothelial cells in the retina were counted. For cone density, numbers of PNA-positive cells were counted within the entire nuclear layer of the field image. The surface of each section was measured on digital microphotographs using the ImageJ v1.4 software. The mean of the density profile measurements across 5 retinal sections that were 100 μm apart per eye was used as a single experimental value. The density profiles were expressed as the mean number of apoptotic cells or cone cells/mm^2^ for 5 samples of each group.

To quantify HNE immunoreactivity, fluorescence intensity was measured on 6–10 randomly selected fields and calculated using the Image J software. The data were extracted and averaged within the group.

### 4.7. Statistical Analysis

We tested the normality of data distribution for IOP, ERG, 4-HNE, PNA, and TUNEL assays using the Shapiro–Wilk test. The data were then analyzed by one-way analysis of variance (ANOVA) and Tukey’s HSD (honestly significant difference) test for multiple pair-wise comparisons. The means and standard error of means (SEM) are reported. α-level was set at 0.05 for statistical significance.

## Figures and Tables

**Figure 1 ijms-24-07362-f001:**
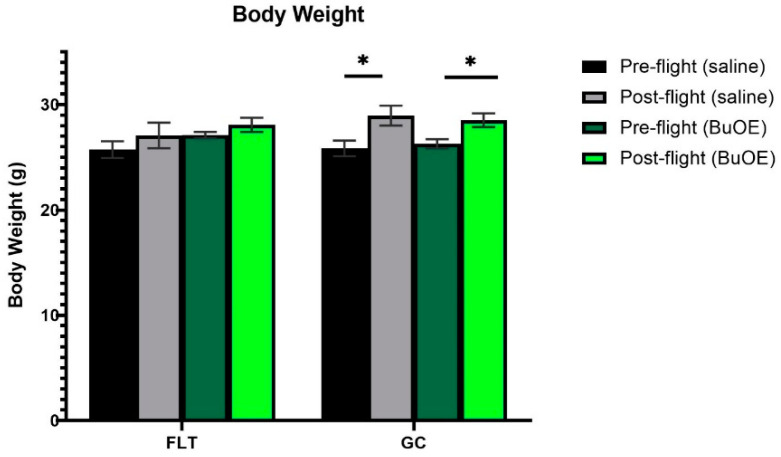
Pre- and post-flight body mass in the flight (FLT) and habitat ground control (GC) groups. Each bar represents the mean ± SEM for mice in each group. * Significantly higher than pre-flight measurements (*p* < 0.05).

**Figure 2 ijms-24-07362-f002:**
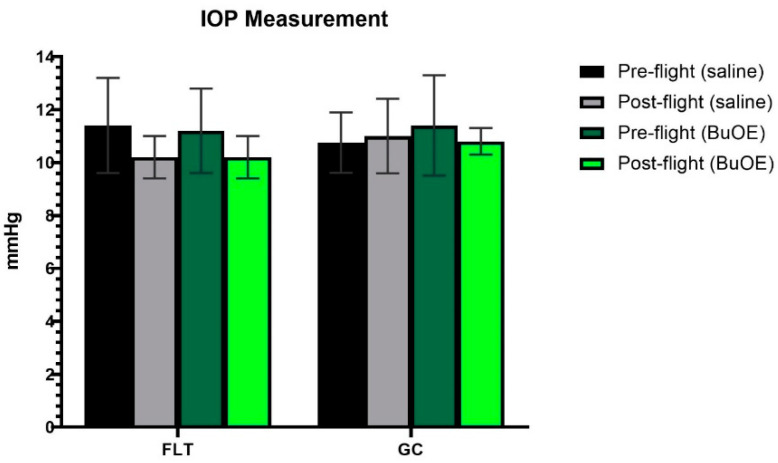
Intraocular pressure (IOP) measurements. Each bar represents the mean counts from the left or right eyes of pre- and post-flight IOP for flight (FLT) and ground control (GC) mice (n = 8). Six readings were made to obtain an average count as a single IOP readout. Values are the means ± SEM.

**Figure 3 ijms-24-07362-f003:**
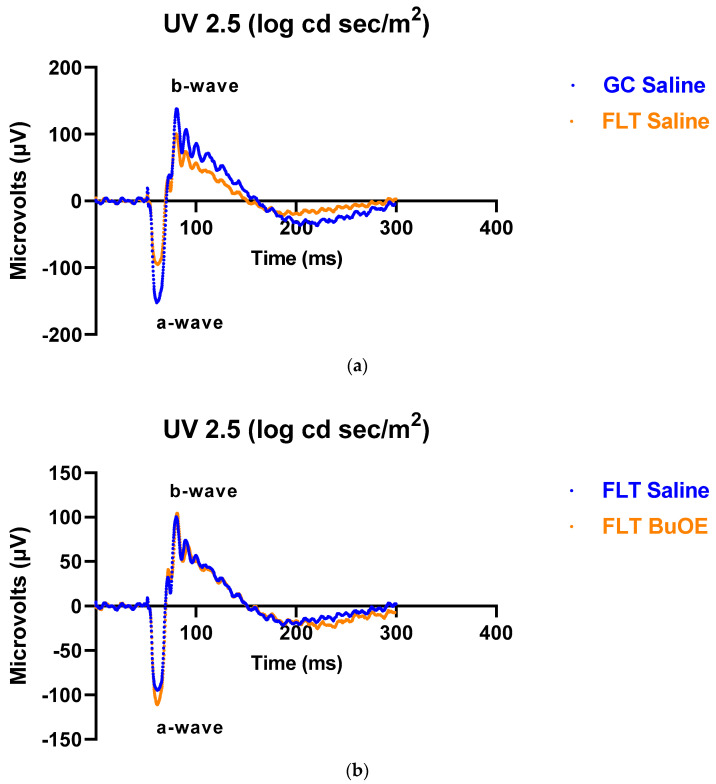
Representative electroretinography (ERG) of the flight (FLT) and habitat control (GC) mice over time. ERGs were obtained in response to the flash intensity of 2.5 log cd·s/m^2^. (**a**) FLT-saline vs. GC-saline. (**b**) FLT-saline vs. FLT-BuOE.

**Figure 4 ijms-24-07362-f004:**
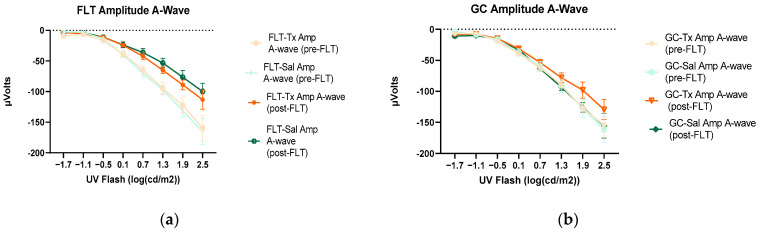
Electroretinography (ERG) a-waves of (**a**) changes in the amplitudes of pre- and post-flight (FLT) groups, and (**b**) changes in the amplitudes of pre- and post-flight ground control (GC) groups in response to increasing flash intensities. (**c**) Changes in the implicit time of post-flight FLT and GC groups. Each point represents the mean ± SEM (n = 8). Note that the response is lower with increased flash intensity in the post-flight group compared to the pre-flight response. The response is similar between pre- and post-flight measurements in the GC group.

**Figure 5 ijms-24-07362-f005:**
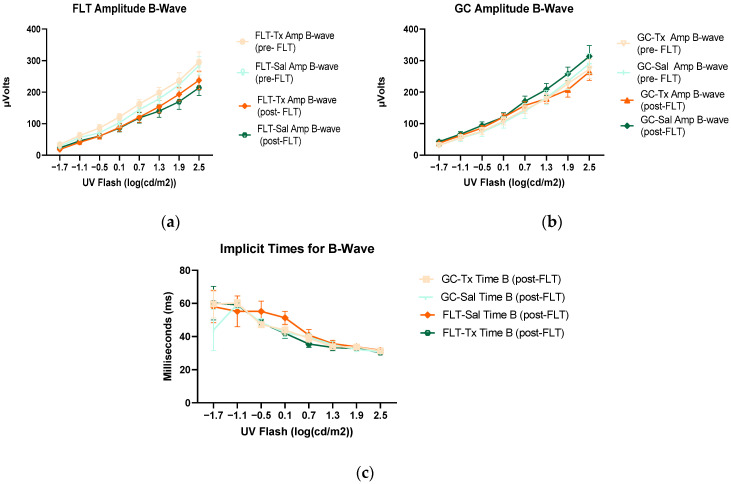
Electroretinography (ERG) b-waves of (**a**) changes in the amplitudes of pre- and post-flight flight (FLT) groups. (**b**) Changes in the amplitudes of pre- and post-flight ground control (GC) groups in response to increasing flash intensities. (**c**) Changes in the implicit time of post-flight FLT and GC groups. Each point represents the mean± SEM (n = 8). Note that the response is lower with increased flash intensity in the post-flight group compared to the pre-flight response. The response is similar between pre- and post-GC measurements.

**Figure 6 ijms-24-07362-f006:**
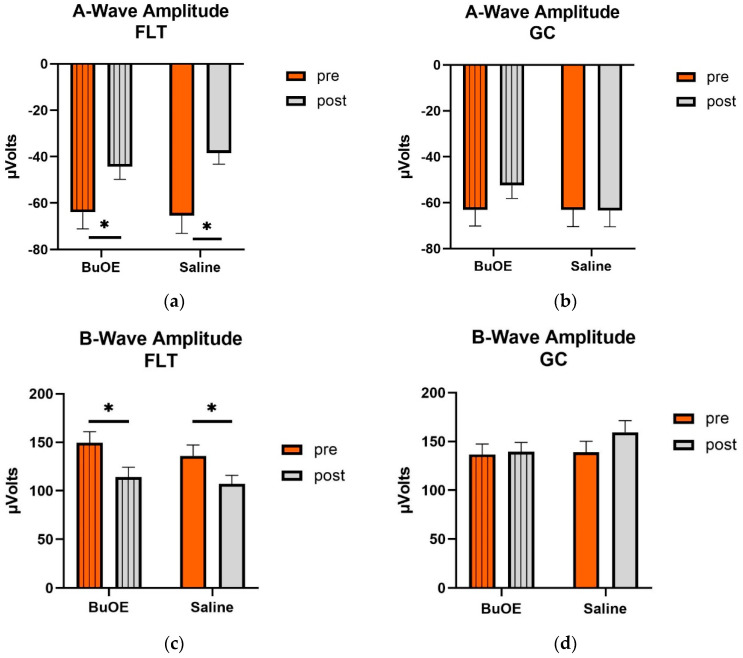
Pre- and post-flight comparison of the average response to the light sensitivity of the retina to the a- and b-wave in saline and BuOE-treated (FLT) and habitat control (GC) groups. The amplitudes were averaged over 8 flash strength from −1.7 to 2.5 log cd s/m^2^. (**a**) The a-wave FLT group. * Significantly different between pre- and post-flight in both saline and BuOE groups (*p* < 0.05). (**b**) The a-wave GC group. No significant difference among groups. (**c**) The b-wave FLT group. * Significantly different between pre- and post-flight in both saline and BuOE groups (*p* < 0.05). (**d**) The b-wave GC group. No significant difference among groups. Bars represent the mean+ SEM (n = 8).

**Figure 7 ijms-24-07362-f007:**
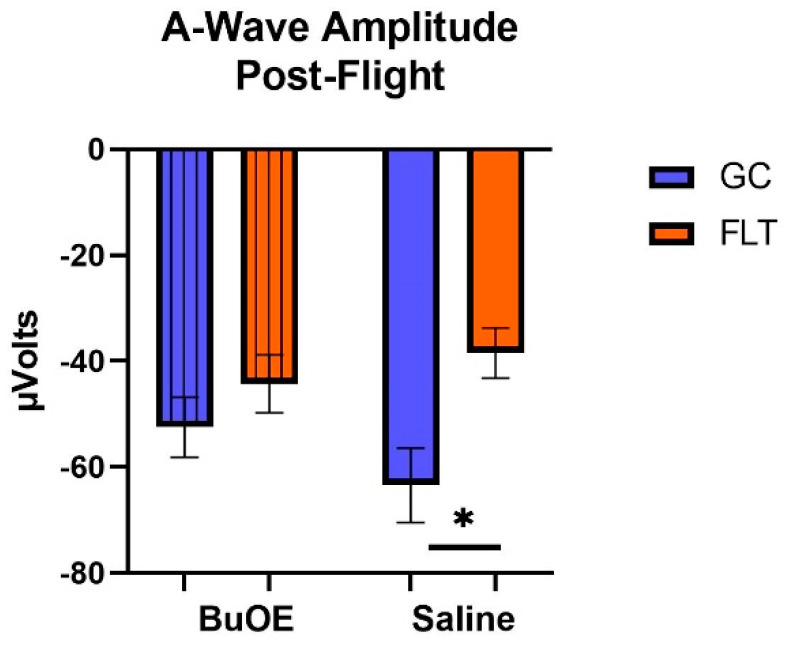
The post-flight comparison of the retina light sensitivity to the a-wave in saline or BuOE-treated flight (FLT) and habitat ground control (GC) groups. * Significantly different between flight and control (*p* < 0.05). Bars represent the mean ± SEM (n = 8).

**Figure 8 ijms-24-07362-f008:**
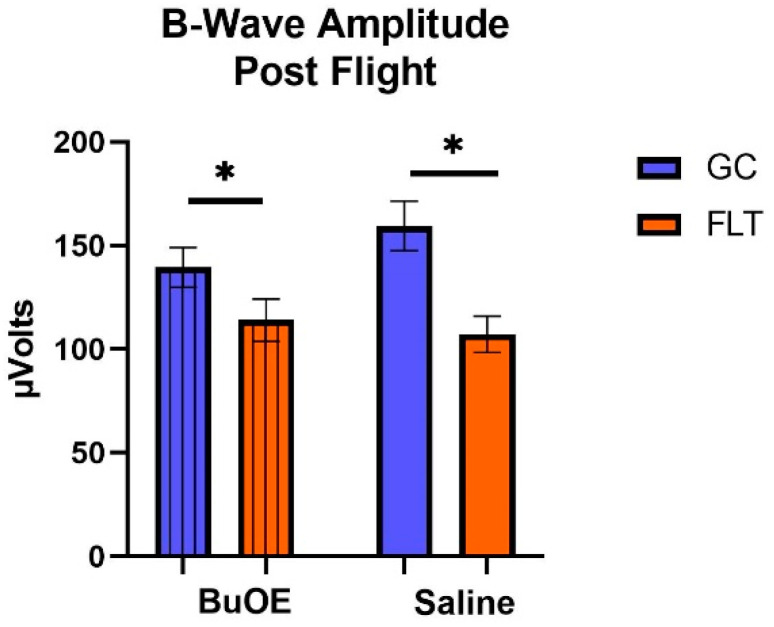
The post-flight comparison of the retina light sensitivity to the b-wave in saline or BuOE-treated flight (FLT) and habitat control (GC) groups. * Significantly different between flight and control (*p* < 0.05). Bars represent the mean ± SEM (n = 8).

**Figure 9 ijms-24-07362-f009:**
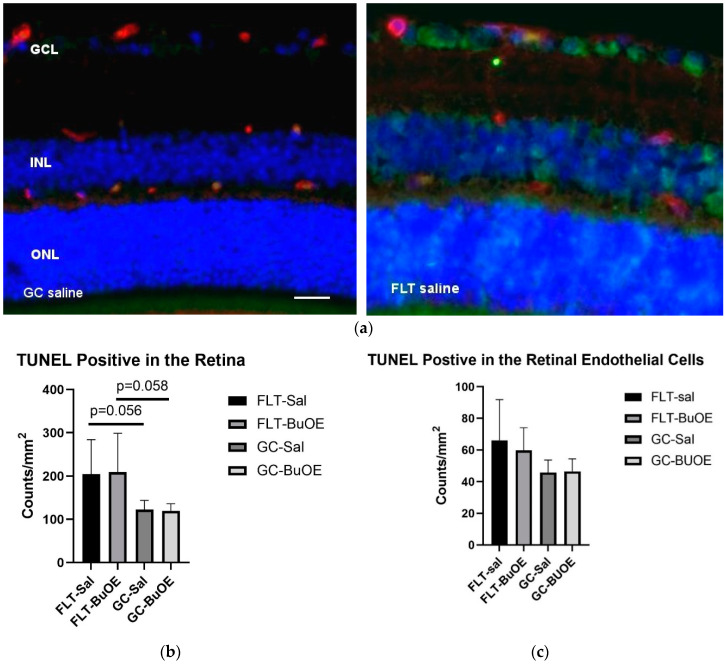
Quantification of apoptosis based on terminal deoxynucleotidyltransferase dUTP nick-end label (TUNEL) staining of flight (FLT) and ground control (GC) mouse retina. (**a**) TUNEL-positive cells were identified with green fluorescence and the endothelium was stained with lectin in red. TUNEL-positive cells that were laid within red lectin-labeled endothelium were identified as TUNEL-positive endothelial cells. The nuclei of photoreceptors were counterstained with DAPI (blue). Scale bar = 50 μm. (**b**) Apoptotic cell density in the retinal outer nuclear layer (ONL), inner nuclear layer (INL), and the ganglion cell layer (GCL) of the FLT and the GC mice. (**c**) Apoptotic cell density in the retinal endothelium of the FLT and GC groups. Values are represented as the mean density ± SEM (n = 5). The density profiles were expressed as the mean number of apoptotic-positive cells/mm^2^. Measurements across five retinal sections per eye were used as a single experimental value. Strong trend differences between FLT saline vs. GC saline (*p* = 0.058), and FLT BuOE vs. GC BuOE (*p* = 0.056).

**Figure 10 ijms-24-07362-f010:**
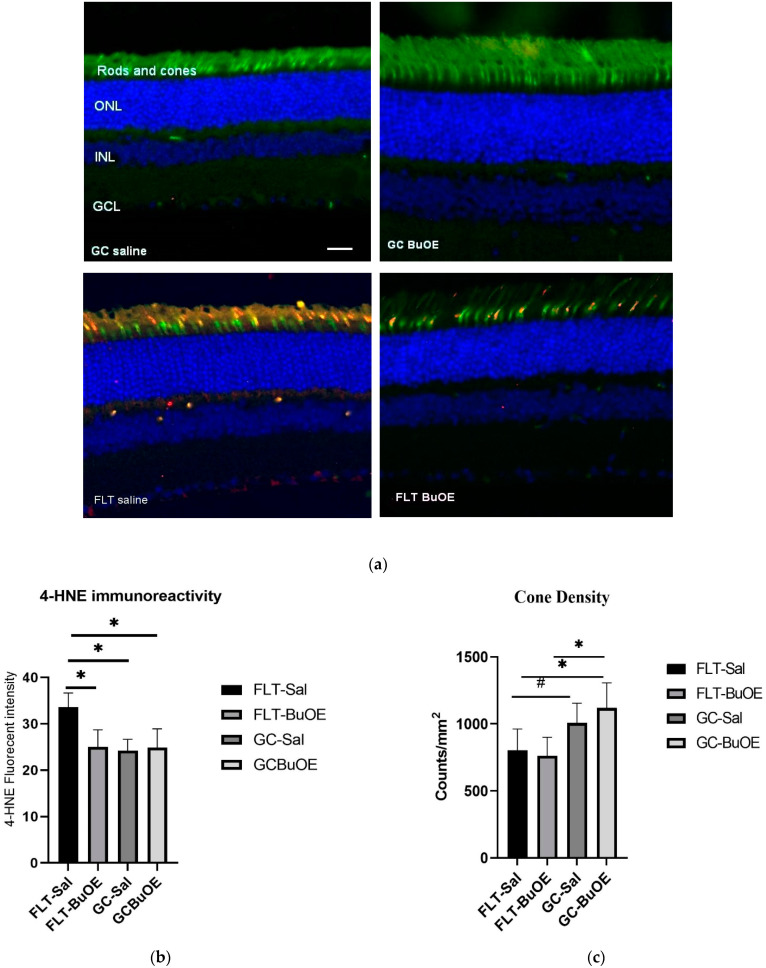
Oxidative damage in the retinal nuclear layer and photoreceptor layer. (**a**) Representative micro-graphs of ocular sections were evaluated in the flight (FLT) and ground control (GC) groups. Immunofluorescence staining for peanut agglutinin (PNA), a marker for cone photoreceptors (green), and 4-hydroxynonenal (4-HNE), a marker for oxidative stress (red), and the nuclei were counterstained with DAPI (blue). Scale bar = 50 μm. (**b**) The average fluorescence intensity of 4-HNE immunoreactivity. Measurements across five retinal sections per eye were used as a single experimental value. Values are represented as the mean density ± SEM (n = 5). * Significance higher in fluorescence intensity of the FLT-saline group compared to all other groups (*p* < 0.05). (**c**) Cell density for PNA-positive cone photoreceptors. * Significance different between FLT-BuOE vs. GC-BuOE or FLT-saline vs. GC-BuOE (*p* < 0.05). # Strong trend difference between FLT-saline and GC-saline (*p* = 0.068).

**Table 1 ijms-24-07362-t001:** Full scope of electroretinogram (ERG) intensity flashes and stimulus intervals.

UV Flash Strength (log(cd/m^2^))	UV Flash Strength (log cd s/m^2^)	Duration (ms)	Delay (s)	Number of Sweeps
1.3	−1.7	1	1	5
1.9	−1.1	1	1	5
2.5	−0.5	1	1	5
3.1	0.1	1	3	5
3.7	0.7	1	5	5
4.3	1.3	1	10	5
4.9	1.9	1	20	5
5.5	2.5	1	20	5

## Data Availability

Data supporting reported results are available upon request. Data will be available at NASA Open Science Data Repository.

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
