# Peer review of "Evidence of Spaceflight-Induced Adverse Effects on Photoreceptors and Retinal Function in the Mouse Eye"

_ijms, 2023, doi:10.3390/ijms24087362_

Round 1

Reviewer 1 Report

The article entitled “Evidence of spaceflight-induced adverse effects on photoreceptor and retinal function in the mouse eye” by Mao and colleagues describes a battery of structural and functional assays performed to measure possible effects of space flight on the visual system. My comments are posted below.

Results, body weight: it seems a bit odd to talk of “pre” and “post” flight for ground control mice, since I guess they were just a bit older. Mabe say “age_matched control”?

This reviewer is very confused by the ERG procedure: why use just UV light? Mice have UV-sensitive cones, but under scotopic conditions these would not be stimulated, at least at lower flash intensities. But rods would not be excited at all, their spectral absorbance curve is far from 360nm. The differences between flight and ground controls are clear, but the authors need to fully explain their reasons for using this protocol, how it would work and what are the interpretations. Furthermore, this would have little bearing on human retinal responses since our shortwave-sensitive cones are blue shifted.

Also very confusing is the section dealing with apoptosis. The authors write: "Apoptotic cells were mostly detected in the retinal ganglion cell layer (GCL) and inner nuclear layer (INL). The data demonstrate that space flight conditions induce apoptosis in retina photoreceptors.“ But there are no images of TUNEL staining, and the two sentences are in contradiction (one speaks of INL/GCL, then about photoreceptor damage)? I see no data showing anything like this. How do they discriminate retinal from EC Staining? They need to include representative images of TUNEL staining in different groups, this is not acceptable as is.

Figure 10 is very poor, PNA staining should be much more pronounced, the resolution is insufficient. Why does rod outer segment stain appear in the upper panels, is this non-specific background? And it is not clear how cone density was measured, typically this needs to be performed on flat-mounted whole retinas.

Finally, the discussion is too vague, there is still no direct proof of photoreceptor damage, as it stands it is impossible to equate reduced ERG responses with an apoptosis.

In conclusion, the aims of the study are interesting and important, it is disappointing that the experimental proofs and explanations are not adequate and need a lot of improvement.

Author Response

The article entitled “Evidence of spaceflight-induced adverse effects on photoreceptor and retinal function in the mouse eye” by Mao and colleagues describes a battery of structural and functional assays performed to measure possible effects of space flight on the visual system. My comments are posted below.

Results, body weight: it seems a bit odd to talk of “pre” and “post” flight for ground control mice, since I guess they were just a bit older. Mabe say “age_matched control”?

Response: We intended to make it clear that control groups were tested and measured on the same schedule as flight groups. We presented the results for pre- vs post-flight flight group as well as pre- and post-flight ground control as a comparison. Ground control groups were housed at exact same NASA hardware as flight group. As suggested reviewer, “age-matched control” is added to GC group to clarify the group assignment.

This reviewer is very confused by the ERG procedure: why use just UV light? Mice have UV-sensitive cones, but under scotopic conditions these would not be stimulated, at least at lower flash intensities. But rods would not be excited at all, their spectral absorbance curve is far from 360nm. The differences between flight and ground controls are clear, but the authors need to fully explain their reasons for using this protocol, how it would work and what are the interpretations. Furthermore, this would have little bearing on human retinal responses since our shortwave-sensitive cones are blue shifted.

Response: We thank reviewer for pointing that out. The mouse retina expresses a short wavelength-sensitive and a middle/long wavelength-sensitive opsin (S- and M-opsin) (1) which the S-cones are responsive to UV (peak 360 nm) while M-cones and rods response to the green light at peak 504 nm. Previous studies suggested that the mouse cone system predominantly expresses S-opsin. Since S-opsin sensitivity drops significantly at wavelengths longer than 425 nm, Only UV light are efficient in stimulating mouse S-opsin (2). Here, in our present study, we focused on probing S- cone-mediated properties using ultraviolet light stimulation. Despite the small percentage of cones in the mouse photoreceptor (5%), cone-mediated responses in mouse ganglion cells are fast and sensitive. Our measurement enables us to quantify cone-mediate response supported by histochemistry evaluation of cone cells. In term of translational value, cone bipolar pathways are similar to those found in higher mammals (3) Therefore, the mouse is potentially a useful model to study cone-mediated vision and diseases that affect cone function in humans (3,4). These info are added to the Discussion. As suggested by reviewer, in the further study a green light with 520 nm will be added.

  1. Nadal-Nicolas, F. M. et al. True S-cones are concentrated in the ventral mouse retina and wired for color detection in the upper visual field. Elife 9, doi:10.7554/eLife.56840 (2020).
  2. Wang, Y. V., Weick, M. & Demb, J. B. Spectral and temporal sensitivity of cone-mediated responses in mouse retinal ganglion cells. J Neurosci 31, 7670-7681, doi:10.1523/JNEUROSCI.0629-11.2011 (2011).
  3. Wassle, H., Puller, C., Muller, F. & Haverkamp, S. Cone contacts, mosaics, and territories of bipolar cells in the mouse retina. J Neurosci 29, 106-117, doi:10.1523/JNEUROSCI.4442-08.2009 (2009).
  4. Nikonov, S. S., Kholodenko, R., Lem, J. & Pugh, E. N., Jr. Physiological features of the S- and M-cone photoreceptors of wild-type mice from single-cell recordings. J Gen Physiol 127, 359-374, doi:10.1085/jgp.200609490 (2006).

Also very confusing is the section dealing with apoptosis. The authors write: "Apoptotic cells were mostly detected in the retinal ganglion cell layer (GCL) and inner nuclear layer (INL). The data demonstrate that space flight conditions induce apoptosis in retina photoreceptors.“ But there are no images of TUNEL staining, and the two sentences are in contradiction (one speaks of INL/GCL, then about photoreceptor damage)? I see no data showing anything like this. How do they discriminate retinal from EC Staining? They need to include representative images of TUNEL staining in different groups, this is not acceptable as is.

Response: We thank reviewer for the comment. The apoptotic cells were detected in GCL and INL layer. Representative images are added to the manuscript. The sentence is revised to clarify the results.

Figure 10 is very poor, PNA staining should be much more pronounced, the resolution is insufficient. Why does rod outer segment stain appear in the upper panels, is this non-specific background? And it is not clear how cone density was measured, typically this needs to be performed on flat-mounted whole retinas.

Response:  Unfortunately, the quality of the images was limited by microscope available to us. We used the BZ-X710 All-in-One inverted fluorescence microscope for this project. However, several better quality images are uploaded with this revision. In the further study, we will seek to use confocal microscope to improve the quality of the images. Yes. The reviewer is right, there is non-specific background stainings. Due to limited tissues from flight study, evaluation using a flat-mounted whole retina is not feasible. For cone density, numbers of PNA-positive cells were counted within the entire nuclear layer. The surface of each section was measured on digital microphotographs using ImageJ v1.4 software. The mean of the density profile measurements across 5 retina sections per eye was used as a single experimental value.

Finally, the discussion is too vague, there is still no direct proof of photoreceptor damage, as it stands it is impossible to equate reduced ERG responses with an apoptosis.

Response: With these data, it is hard to draw conclusion of direct association of cellular damage and functional change assessed by ERG. More comprehensive testing including electro-oculograph, visual evoked potentials, and physiologic/pathophysiologic evaluation related to retinal function will be performed in our further study to test the cone and rod response. The associated between percent of cell loss and cell type that lead to functional alteration will be studied with evaluation for various cell type in the retina. We aim to define a threshold for percentage loss of cells below which retinal dysfunction will occur. Study has shown the impact of disruption of BRB function on retinal function. The biomarker for blood-retinal barrier integrity and function will also be tested.

Reviewer 2 Report

The counts for apoptosis were similar between the BuOE-treated and saline flight groups. Apoptotic cells were mostly detected in the retinal ganglion cell layer (GCL) and inner nuclear layer (INL). The data demonstrate that space flight conditions induce apoptosis in retina photoreceptors.

This result needs to be corrected. Either apoptosis was in the GCL and INL or in the photoreceptor layer.

Author Response

The counts for apoptosis were similar between the BuOE-treated and saline flight groups. Apoptotic cells were mostly detected in the retinal ganglion cell layer (GCL) and inner nuclear layer (INL). The data demonstrate that space flight conditions induce apoptosis in retina photoreceptors.

This result needs to be corrected. Either apoptosis was in the GCL and INL or in the photoreceptor layer.

Response: We thank the reviewer for pointing that out. The sentence is revised to clarify the statement that apoptotic cell death was mostly found in the GCL and INL. Representative images are added to demonstrate these findings.

Reviewer 3 Report

Spaceflight Associated Neuro-ocular Syndrome (SANS) is ophthalmic abnormality that astronauts experience after they returned to Earth. The causes of this condition are not known.  One possible explanation is the radiation induced oxidative stress which has been known to cause retinal damage and retinal cell death. The purpose of the present study is to investigate retinal structure, function, and oxidative stress in retina after space flight. They also evaluated the efficacy of SOD like compound, BuOE, in these parameters. Male mice were sent to ISS for 35 days and their visual function was evaluated at 20 hours after the splash down. Space flight inhibited normal weight gain, decreased retinal function, increased apoptotic cells and lipid peroxidation in photoreceptors and decrease cone viability. Space flight did not altered IOP. Authors also found that BuOE prevented lipid peroxidation in the mice photoreceptors. They concluded space flight induce oxidative stress which may alter retinal function and  photoreceptors cell viability. They also plan to further investigate the effects are transient by evaluating retina from 3 months after the space flight. I think the study is unique and interesting to many audience. Below are my comments and suggestions,

11.       Details of mice rearing environment should be well described e.g. temperature, humidity, light intensity, cage size, water, air or any condition may be different in FLT mice from GC mice.

22.       I believe the body weight data does not add to the authors conclusion, but if authors decide to add, authors may describe mice activity level (a running wheel or not?) and food consumption. Gravity change may reduce energy expenditure in mice which may have inhibited the normal weight gain in mice.

33.       Authors may describe a lighting cycle condition (e.g. 12 hours light: 12 hours dark) that FLT and CG mice were exposed. ISS revolves around Erath in 90 min.  Several evidence indicates disruption of circadian rhythms modulate retinal function and structures.

44.       Authors may explain why they used 360 nm light instead of 504 nm which may be more suitable for dark adapted ERG.

55.       Authors should re-plot representative ERG traces to make figures more publication quality. They seem to be a copy/paste image from MS excel. Boarders and lines should be deleted. Time scales should be bottom of the x axis. “UV 2.5” should be mentioned in figure legends. In figure 3B, there is a fragment “a-“

66.       For Figure 7b and 8b, I do not see any meaning to show these data.

77.       In line 150, authors mentioned that the apoptotic cells were mostly in GCL and INL. Author should add data in figures.  

88.       For Figure 9a and 10c, author indicated the mark as a strong trend. Trend is subjective and unambiguous term. Trend should not be marked to avoid confusion.

99.       For figure 10, authors should describe how fluorescence intensity and cone counts were measured e.g. count of an entire retina, in central or peripheral retina, every 500 um in retina etc.

110.   Authors also should show most representative images in Figure 10. I do not see 4HNE staining in INL and GCL in FLT reina.

111.   Authors may be suggested to measure thickness of ONL and INL or DAPI cell counts (it could indicate rod viability since only 4% are cones in ONL) in these area to show another parameter of retinal structure.

112.   BuOE is the SOD analog which supposes to turn superoxide to hydrogen peroxide. The process of lipid peroxidation is normally mediated via hydroxyl radical.  Authors may explain the possible mechanism for the BuOE prevention of lipid peroxidation in cone photoreceptors.

Author Response

Details of mice rearing environment should be well described e.g. temperature, humidity, light intensity, cage size, water, air or any condition may be different in FLT mice from GC mice.

Response: Ground control (GC) mice were housed at the same housing condition, temperature, CO2, food and water as simulation control for flight group. All mice were maintained at an ambient temperature of 26–28 °C with a 12-hour light/dark cycle during the flight. The hardware has a housing density that is within the guidelines recommended by the National Institutes of Health.  All mice were provided NASA Nutrient-upgraded Rodent Food Bar (NuRFB) and autoclaved deionized water ad libitum for nourishment. These info are added to the section.

 Further details regarding housing and other environmental conditions are described at the NASA website https://nlsp.nasa.gov/view/lsdapub/lsda_hardware/IDP-LSDA_HARDWARE-0000000000000523..

          I believe the body weight data does not add to the authors conclusion, but if authors decide to add, authors may describe mice activity level (a running wheel or not?) and food consumption. Gravity change may reduce energy expenditure in mice which may have inhibited the normal weight gain in mice.

Response: Discussion regarding body weight differences between flight and ground control groups are added to the discussion section as followed: Our results show that FLT mice did not gain weight after the flight as was observed for age-matched GC group.  Other spaceflight studies reported a loss in body mass after spaceflight. It has been proposed that the lack of increase in body mass may be related to mixed factors including dehydration, stress, altered glucose, and lipid metabolism, or less food consumption.

Authors may describe a lighting cycle condition (e.g. 12 hours light: 12 hours dark) that FLT and CG mice were exposed. ISS revolves around Erath in 90 min.  Several evidence indicates disruption of circadian rhythms modulate retinal function and structures.

Response: Both flight and ground controls were under 12-hour light/dark cycle during the flight.  

         Authors may explain why they used 360 nm light instead of 504 nm which may be more suitable for dark adapted ERG.

Response: The mouse retina expresses a short wavelength-sensitive and a middle/long wavelength-sensitive opsin (S- and M-opsin) (1) which the S-cones are responsive to UV (peak 360 nm) while M-cones and rods response to green light at peak 504 nm. Previous studies suggested that the mouse cone system predominantly expresses S-opsin. Since S-opsin sensitivity drops significantly at wavelengths longer than 425 nm, Only UV light are efficient in stimulating mouse S-opsin (2). Here, in our present study, we focused on probe S- cone-mediated properties using ultraviolet light stimulation. Despite the small percentage of cones in the mouse photoreceptor (5%) (3), cone-mediated responses in mouse ganglion cells are fast and sensitive. Our measurement enables us to quantify cone-mediate response supported by histochemistry evaluation. These info are added in the discussion.

  1. Nadal-Nicolas, F. M. et al. True S-cones are concentrated in the ventral mouse retina and wired for color detection in the upper visual field. Elife 9, doi:10.7554/eLife.56840 (2020).
  2. Wang, Y. V., Weick, M. & Demb, J. B. Spectral and temporal sensitivity of cone-mediated responses in mouse retinal ganglion cells. J Neurosci 31, 7670-7681, doi:10.1523/JNEUROSCI.0629-11.2011 (2011).
  3. Nikonov, S. S., Kholodenko, R., Lem, J. & Pugh, E. N., Jr. Physiological features of the S- and M-cone photoreceptors of wild-type mice from single-cell recordings. J Gen Physiol 127, 359-374, doi:10.1085/jgp.200609490 (2006).

       Authors should re-plot representative ERG traces to make figures more publication quality. They seem to be a copy/paste image from MS excel. Boarders and lines should be deleted. Time scales should be bottom of the x axis. “UV 2.5” should be mentioned in figure legends. In figure 3B, there is a fragment “a-“

Response: We thank the reviewer for pointing that out. The figures are revised as suggested by reviewer.

        For Figure 7b and 8b, I do not see any meaning to show these data.

Response: As suggested by reviewer, these two figures are deleted.

In line 150, authors mentioned that the apoptotic cells were mostly in GCL and INL. Author should add data in figures. 

Response: Figures are added to show apoptosis in the GCL and INL.

For Figure 9a and 10c, author indicated the mark as a strong trend. Trend is subjective and unambiguous term. Trend should not be marked to avoid confusion.

Response: Because of the uniqueness of the flight study, we would like to include the data that may indicate flight-induced physiological changes. Studies have shown that flight mouse showed great variations within the group compared to controls. Significant change may occur if we are able to increase group size. However, if reviewer insists, we will remove mark for trend.

For figure 10, authors should describe how fluorescence intensity and cone counts were measured e.g. count of an entire retina, in central or peripheral retina, every 500 um in retina etc.

Response: More detailed descriptions regarding evaluation of fluorescence intensity and cone are added to the method. The entire retina is counted. The section was selected for one biomarker staining at every 100 mm apart.

 Authors also should show most representative images in Figure 10. I do not see 4HNE staining in INL and GCL in FLT reina.

Response: The figures are replaced with images that shows HNE positive in the GCL and INL layer.

Authors may be suggested to measure thickness of ONL and INL or DAPI cell counts (it could indicate rod viability since only 4% are cones in ONL) in these area to show another parameter of retinal structure.

Response: Good suggestion. These mice were euthanized within 20 hours of landing. Significant structure changes or cell loss may not be evident. However, we will incorporate these measurements in the long-term time point measurement.

         BuOE is the SOD analog which supposes to turn superoxide to hydrogen peroxide. The process of lipid peroxidation is normally mediated via hydroxyl radical.  Authors may explain the possible mechanism for the BuOE prevention of lipid peroxidation in cone photoreceptors.

Response: Possible mechanism for the BuOE is added in the Discussion Section as suggested by reviewer as follow: Metalloporphyrin catalytic antioxidants are potent detoxifiers of lipid peroxides and lipoperoxyl radical, a process that is mediated via hydroxyl radical. Our present study suggests oxidative stress is a contributing mechanism to observed damage in the retina and photoreceptor. It was demonstrated by increased 4-HNE staining, a biomarker for lipid peroxidation. Adding BuOE, a lipophilic metalloporphyrin compound with high potency of catalyst of lipid peroxidation reduced immunoreactivity of the biomarker in the BuOE-treated ocular tissue compared to flight saline group, suggesting a potential mechanism of oxidative stress in inducing cellular damage.

Round 2

Reviewer 1 Report

The authors have improved the manuscript, but there are still several remaining questions that need answering.

The ERG data are still a bit confusing. Why do the authors speak of « scotopic » ERGs when they admit they are just measuring S cone activity? Scotopic recording is only done to get data on rod activity. They should also provide implicit time data in figures 4 and 5, since these should align with the amplitude effects (ie. one could expect increased latencies if cone function is compromised). Which stimulus intensity are they using to report significant differences (Fig. 6), I guess it is 2.5 cd but this is not stated. The figure 7 showing a protective effect of BuOE on the a wave value is in contradiction to earlier figures where there is no effect (figures 3A, 6A)?

They cannot get any information on ganglion cell activity by these measurements, especially as mice ERGs do not exhibit i waves as seen in human visual responses (and which are thought to represent ganglion cell activity). I still think the relevance to humans is weakened by focusing on UV cones, as these do not exist in human retinas – although I understand their more general argument that underlying cone circuitry might be similar.

The apoptosis data are also confusing. I had to look many times at their figures before realizing TUNEL label was in green and endothelial marker in red (this detail is not provided !). Even so the labelling appears strange, diffuse within the cytoplasm but I don’t see TUNEL positive nuclei within the nuclear layers in new figure 9, the distribution of staining is compatible with blood vessels in the inner retina. So I really do not see how they attribute such high counts to the retina (figure 9b, c)?

Finally, there is also a problem with the cone density estimates. They have used multiple thin sections to estimate cone numbers, which is notoriously poor for doing such counts (M M LaVail  1 M T MatthesD YasumuraR H Steinberg Variability in rate of cone degeneration in the retinal degeneration (rd/rd) mouse Exp Eye Res 1997 65(1):45-50. doi: 10.1006/exer.1997.0308.). They also give their estimates in mm2, which is impossible if they used sections. The difference is not significant (they provide no hard data on means +/-S.D.), so they cannot claim any real effects of space flight on cone numbers.

All these points should addressed in detail.

Author Response

The ERG data are still a bit confusing. Why do the authors speak of « scotopic » ERGs when they admit they are just measuring S cone activity? Scotopic recording is only done to get data on rod activity. They should also provide implicit time data in figures 4 and 5, since these should align with the amplitude effects (ie. one could expect increased latencies if cone function is compromised). Which stimulus intensity are they using to report significant differences (Fig. 6), I guess it is 2.5 cd but this is not stated. The figure 7 showing a protective effect of BuOE on the a wave value is in contradiction to earlier figures where there is no effect (figures 3A, 6A)?

Response: we agree with reviewer’s comment regarding scotopic ERG. The word “scotopic” is removed.

As suggested by reviewer, figures for implicit time over increases flash intensities are added to Figures 4 and 5.

For Figure 6, the data is reported on average response of amplitudes to all flash intensity listed in table 1 (from -1.7 to 2.5 log cd sec/m2). This information is added in the results section for Figure 6 and in the figure legend.

In figure 7,  I do not think we can directly conclude that there is a protective effect by BuOE, since there is no differences between post-flight saline vs post-flight BuOE. Although the amplitude for GC-BuOE and flight-BuOE is similar, it can not be translated to protective effect of BuOE. This result may attribute to differences between GC-saline and GC-BuOE, although there was no significant differences between these two groups.

They cannot get any information on ganglion cell activity by these measurements, especially as mice ERGs do not exhibit i waves as seen in human visual responses (and which are thought to represent ganglion cell activity). I still think the relevance to humans is weakened by focusing on UV cones, as these do not exist in human retinas – although I understand their more general argument that underlying cone circuitry might be similar.

Response: we agree with reviewer. To add translational value, in the further study, a green light with 520 nm will be added.

The apoptosis data are also confusing. I had to look many times at their figures before realizing TUNEL label was in green and endothelial marker in red (this detail is not provided !). Even so the labelling appears strange, diffuse within the cytoplasm but I don’t see TUNEL positive nuclei within the nuclear layers in new figure 9, the distribution of staining is compatible with blood vessels in the inner retina. So I really do not see how they attribute such high counts to the retina (figure 9b, c)?

Response: The labeling info is added in the results and figure legend. The image was readjusted in term of contrast and brightness to better show positive nuclei: double stained nuclei with green(apoptotic cell)  and blue (DAPI nuclei) which mostly concentrated in INL and GCL.

Finally, there is also a problem with the cone density estimates. They have used multiple thin sections to estimate cone numbers, which is notoriously poor for doing such counts (M M LaVail  1 , M T Matthes, D Yasumura, R H Steinberg Variability in rate of cone degeneration in the retinal degeneration (rd/rd) mouse Exp Eye Res 1997 65(1):45-50. doi: 10.1006/exer.1997.0308.). They also give their estimates in mm2, which is impossible if they used sections. The difference is not significant (they provide no hard data on means +/-S.D.), so they cannot claim any real effects of space flight on cone numbers.

Response: The numbers of PNA-positive cells were counted within the surface of each section that was measured on digital microphotographs using ImageJ v1.4 software. Then area measurement was converted from pixel ^2 to mm^2. We understand this counting method is not ideal for quantification. In the future, we plan to use 50 micron sections with images from two-photon microscope for quantification using unbiased stereological analysis method or as suggested by reviewer, we will explore the possibility to use retinal wholemount for evaluation. However, if reviewer insist, we will remove Figure 10c.

Round 3

Reviewer 1 Report

Okay, even though some replies are not really satisfactory I realise you have made a good-faith effort to respond to the best of your abilities and given the constraints. For future studies I still recommend you quantify cones by sampling whole flat-mounted specimens, rather than any section-based approach which will always lead to huge variations. No more comments.